# Biotechnological and Ecological Potential of *Micromonospora provocatoris* sp. nov., a Gifted Strain Isolated from the Challenger Deep of the Mariana Trench

**DOI:** 10.3390/md19050243

**Published:** 2021-04-25

**Authors:** Wael M. Abdel-Mageed, Lamya H. Al-Wahaibi, Burhan Lehri, Muneera S. M. Al-Saleem, Michael Goodfellow, Ali B. Kusuma, Imen Nouioui, Hariadi Soleh, Wasu Pathom-Aree, Marcel Jaspars, Andrey V. Karlyshev

**Affiliations:** 1Department of Pharmacognosy, College of Pharmacy, King Saud University, P.O. Box 2457, Riyadh 11451, Saudi Arabia; wabdelmageed@ksu.edu.sa; 2Department of Pharmacognosy, Faculty of Pharmacy, Assiut University, Assiut 71526, Egypt; 3Department of Chemistry, Science College, Princess Nourah Bint Abdulrahman University, Riyadh 11671, Saudi Arabia; lhalwahaibi@pnu.edu.sa (L.H.A.-W.); msalsaleem@pnu.edu.sa (M.S.M.A.-S.); 4School of Life Sciences Pharmacy and Chemistry, Faculty of Science, Engineering and Computing, Kingston University London, Penrhyn Road, Kingston upon Thames KT1 2EE, UK; b.lehri@hotmail.co.uk; 5School of Natural and Environmental Sciences, Newcastle University, Newcastle upon Tyne NE1 7RU, UK; Michael.Goodfellow@newcastle.ac.uk (M.G.); ali.budhi.kusuma@uts.ac.id (A.B.K.); ino20@dsmz.de (I.N.); hariadi.soleh@pom.go.id (H.S.); wasu.p@cmu.ac.th (W.P.-A.); 6Indonesian Centre for Extremophile Bioresources and Biotechnology (ICEBB), Faculty of Biotechnology, Sumbawa University of Technology, Sumbawa Besar 84371, Indonesia; 7Leibniz-Institut DSMZ—German Collection of Microorganisms and Cell Cultures, Inhoffenstraße 7B, 38124 Braunschweig, Germany; 8Marine Biodiscovery Centre, Department of Chemistry, University of Aberdeen, Old Aberdeen AB24 3UE, UK; m.jaspars@abdn.ac.uk

**Keywords:** Mariana Trench, *Micromonospora provocatoris* MT25, desferrioxamine, *n*-acetylglutaminyl glutamine amide, ^1^H-^15^N 2D-NMR, genomics, biosynthetic gene clusters, stress genes

## Abstract

A *Micromonospora* strain, isolate MT25^T^, was recovered from a sediment collected from the Challenger Deep of the Mariana Trench using a selective isolation procedure. The isolate produced two major metabolites, *n*-acetylglutaminyl glutamine amide and desferrioxamine B, the chemical structures of which were determined using 1D and 2D-NMR, including ^1^H-^15^N HSQC and ^1^H-^15^N HMBC 2D-NMR, as well as high resolution MS. A whole genome sequence of the strain showed the presence of ten natural product-biosynthetic gene clusters, including one responsible for the biosynthesis of desferrioxamine B. Whilst 16S rRNA gene sequence analyses showed that the isolate was most closely related to the type strain of *Micromonospora chalcea*, a whole genome sequence analysis revealed it to be most closely related to *Micromonospora tulbaghiae* 45142^T^. The two strains were distinguished using a combination of genomic and phenotypic features. Based on these data, it is proposed that strain MT25^T^ (NCIMB 15245^T^, TISTR 2834^T^) be classified as *Micromonospora provocatoris* sp. nov. Analysis of the genome sequence of strain MT25^T^ (genome size 6.1 Mbp) revealed genes predicted to responsible for its adaptation to extreme environmental conditions that prevail in deep-sea sediments.

## 1. Introduction

Novel filamentous actinobacteria isolated from marine sediments are a prolific source of new specialized metabolites [1,2,3], as examplified by the discovery of the abyssomicins, a new family of polyketides [4] produced by *Micromonospora* (formerly *Verrucosispora*) *maris* [5] and the proximicins, novel aminofuran antibiotics and anticancer compounds isolated from *Micromonospora* (*Verrucosispora*) *fiedleri* [6,7]. Novel micromonosporae have large genomes (6.1–7.9 Mbp) which contain strain, species and clade specific biosynthetic gene clusters (BGCs) with the potential to express new bioactives [8,9,10] needed to counter multi-drug resistant pathogens [11]. These developments provide an objective way of prioritizing novel micromonosporae for genome mining and natural product discovery [8,9,10]. Key stress genes detected in the genomes of micromonosporae provide an insight into how they became adapted to harsh abiotic conditions that are characteristic of extreme biomes [8,12].

The actinobacterial genus *Micromonospora* [13] emend Nouioui et al. [5], the type genus of the family *Micromonosporaceae* [14] emend Nouioui et al. [5] is a member of the order *Micromonosporales* [15] of the class *Actinomycetia* [16]. The genus encompasses 88 validity published species (www.bacterio.net.micromonospora, accessed on 28 May 2018), including the type species *Micromonospora chalcea* [13,17]. *Micromonospora* species can be distinguished using combinations of phenotypic properties [10,18]. The application of cutting-edge taxonomic methods showed the genus to be monophyletic, clarified its subgeneric structure and provided a sound framework for the recognition of new species [5,8]. The genus typically contains aerobic to microaerophilic, Gram-positive, acid-fast-negative actinobacteria, which form single, nonmotile spores on an extensively branched substrate mycelium, lack aerial hyphae and produce either xylose or mannose or galactose and glucose as major sugars. Hydrolysates of these microorganisms are rich in *meso*- and/or dihydroxypimelic acid (A_2_pm) with phosphatidylethanolamine being a diagnostic polar lipid. Iso-C15:0 and iso-C16:0 are the predominant fatty acids, while the DNA G + C percentage ranges from 65% to 75% [5,8,16].

The present study was designed to determine the taxonomic status, biotechnological potential and ecological characteristics of a *Micromonospora* strain, isolate MT25^T^, recovered from sediment collected from the Challenger Deep of the Mariana Trench in the Pacific Ocean. The results of the polyphasic study together with associated genomic features showed that isolate MT25^T^ represents a novel species within the genus *Micromonospora* and has a large genome with the potential to express new natural products, as well as stress-related genes that provide an insight into its ability to tolerate extreme environmental conditions found in deep-sea sediments.

## 2. Results and Discussion

In this study we sequenced, annotated and analyzed the genome of isolate MT25^T^ which was recovered from sediment collected at a depth of 10,898 m from the Mariana Trench, to highlight on its taxonomic status, ability to synthesize major metabolites, as well as giving an insight into its ecological properties.

### 2.1. Isolation, Maintenance and Characterization of Strain MT25^T^

*Micromonospora* strain MT25^T^ was isolated from a sediment sample (no. 281) taken from the Mariana Trench (Challenger Deep; 142°12′372′′ E; 11°19′911′′ N) using a standard dilution plate procedure [19] and raffinose-histidine agar as the selective medium [20]. The sediment was collected at a depth of 10,898 m by the remotely operated submersible Kaiko, using a sterilized mud sampler during dive 74 [21]. The sample (approximately 2 mL) was taken to the UK in an isolated container at 4 °C, then stored at −20 °C.

The strain is aerobic, Gram-positive, nonmotile, produces a branched substrate mycelium bearing single sessile spores with rugose surfaces (0.8–0.9 µm) (Figure 1 and Appendix A). *Meso*-A2pm is the diamino acid of the peptidoglycan, and glucose, mannose, ribose and xylose are the major whole organism sugars. Iso-C16:0 is the predominant fatty acid, and the polar lipids are diphosphatidylglycerol, phosphatidylethanolamine (diagnostic component), phosphatidylinositol, a glycolipid and two unknown phospholipids (Appendix A). Like the other micromonosporae, the strain contains complex mixtures of saturated and unsaturated fatty acids [8,16], as shown by the presence of major proportions of iso-C16:0 (25.3% of total), anteiso-C15:OH (10.5%) and iso-C15:0 (10.0%), lower proportions of iso-C14:0 (3.6%), anteiso-C15:0 (6.9%), C16:0 (2.8%), 10-methyl C16:0 (3.0%), iso-C17:0 (4.2%), anteiso-C17:0 (8.6%), C17:1w9c (3.2%), C17:0 (3.2%), 10-methyl C17:0 (2.1%), iso-C18:0 (1.4%), C18:1w9c (3.5%), C18:0 (4.8%), 10-methyl C18:0 (1.2%) and trace amounts (<1.0%) of iso-C10:0 (0.1%), C10:0 2OH (0.1%), iso-C12:0 (0.1%), C12:0 (0.2%), iso-C13:0 (0.1%), anteiso-C13:0 (0.1%), C14:0 (0.7%), iso-C16:1 (0.6%), iso-C16:1w9c (0.6%), iso-C15:0 3OH (0.5%), anteiso-C17:1 (0.5%), iso-C17:0 3OH (0.1%) and iso-C19:0 (0.1%).

### 2.2. Compound Identification

Compound **1** was obtained as a white amorphous powder, 16.2 mg; [α]_D_^25^= −47 (*c* 0.1, MeOH). The IR absorption peaks of **1** suggested NH_2_ (3408, 3326, 3316, 3274, 3230, 3202 cm^−1^) and carbonyl groups (1670, 1660, 1655, 1647). LRESIMS measurements revealed peaks at *m*/*z* 316.1 [M + H]^+^ and 338.1 [M + Na]^+^ indicating that the molecular weight was 315.1. The molecular formula of **1** was established as C_12_H_21_N_5_O_5_ by HRESIMS (obsd. [M+Na]^+^ at *m*/*z* 338.143098, calcd. for C_12_H_21_N_5_O_5_Na, 338.144046, ∆ = −2.8 ppm), indicating that the molecule had five degrees of unsaturation (Figure 2).

The 1D ^1^H and ^13^C NMR data (DMSO-*d*_6_) in combination with the ^1^H-^13^C HSQC NMR experiments of **1** showed two methine H-5 [*δ_H_* 4.15 (1H, m)] and H-8 [*δ_H_* 4.12 (1H, m)]; four methylene H_2_-10 [*δ_H_* 2.08 (2H, m)], H_2_-3 [*δ_H_* 2.06 (2H, m)], H_2_-4 [*δ_H_* 1.89 (1H, m) and 1.74 (1H, m)], H_2_-9 [*δ_H_* 1.87 (1H, m) and 1.67 (1H, m)] groups and one methyl group CH_3_-15 [*δ_H_* 1.86 (3H, s)] as well as five quaternary carbonyl carbon atoms; C-11 (*δ_C_* 173.8 s), C-2 (*δ_C_* 173.8 s), C-16 (*δ_C_* 173.3 s), C-6 (*δ_C_* 171.4 s) and C-14 (*δ_C_* 169.7 s) (Table 1).

Eight hydrogen resonances lacked correlations in the ^1^H-^13^C HSQC 2D NMR spectrum of **1** and were therefore recognized as being located on either oxygen or nitrogen atoms. The ^1^H-^15^N HSQC NMR spectrum (Appendix A) of **1** indicated that all eight of these protons were bonded to nitrogen (Table 1); three as part of NH_2_ groups; NH_2_-17 [*δ_H_* 7.27 (1H, brs) and 7.05 (1H, brs)], NH_2_-12 [*δ_H_* 7.28 (1H, brs) and 6.75 (1H, brs)], NH_2_-1 [*δ_H_* 7.28 (1H, brs) and 6.75 (1H, brs)] and two in NH groups; NH-13 [*δ_H_* 8.11 (1H, d, 7.8 Hz)], NH-7 [*δ_H_* 7.97 (1H, d, 7.8 Hz)]. Also, from the ^1^H-^15^N HSQC and the ^1^H-^15^N HMBC 2D-NMR spectra (DMSO-*d*_6_) of **1** it was possible to assign the resonance of each nitrogen; NH-13 (*δ_N_* 123.2 t), NH-7 (*δ_N_* 117.3 d), NH_2_-12 (*δ_N_* 108.6 t), NH_2_-17(*δ_N_* 104.8 t) and NH_2_-1 (*δ_N_* 108.5 t) (Figure 3 and Appendix A).

With all protons assigned to their directly bonded carbon and nitrogen atoms it was possible to deduce substructures with the aid of the ^1^H-^1^H COSY spectrum of **1** (Figure 3). The connectivities between substructures were established from key ^1^H-^13^C HMBC correlations (Figure 3). Thus, correlations between C-2 (*δ_C_* 173.8) and H_2_-3, H_2_-1 and between C-11 (*δ_C_* 173.8) and H_2_-10, H_2_-12 as well as between C-5 (*δ_C_* 52.7) and H_2_-3 and between C-8 (*δ_C_* 52.1) and H_2_-17 and between C-14 (*δ_C_* 169.7) and H-5, H-13, H_3_-15 and between C-16 and H-8, H_2_-9 and H_2_-17 clearly defined the planar structure as shown in **1**. Finally, the positions of nitrogen were defined from ^1^H-^15^N-HMBC which showed long range correlations between N-13 and H4a/b and H3-15 and between N-7 and H9a/b (Figure 3 and Appendix A). Given these results and comparisons with previously data [22], the compound was identified as *n*-acetylglutaminyl glutamine amide.

Compound **2** was identified as deferoxamine B. Its molecular formula was established as C_25_H_49_N_6_O_8_ by HRESIMS (*m*/*z* 561.3577 [M + H]^+^, calcd. for C_25_H_49_N_6_O_8_, 561.3592, ∆ = −2.6 ppm), which required five degrees of unsaturation, and also found bound to Fe^+3^ (*m*/*z* 614.2780 [M + Fe − 2H]^+^) (Figure 2).

The full planar structure of **2** was assigned by interpretation of 1D (^1^H and ^13^C) in connection with extensive 2D-NMR (^1^H-^1^H COSY, ^1^H-^13^C HSQC, ^1^H-^13^C HMBC, HSQC-TOCSY and 1,1-ADEQUATE) spectroscopic data recorded in (DMSO-*d*_6_) (Table 1), and by comparing it with the previously reported data on desferrioxamine [23].

The 1D ^1^H and ^13^C NMR spectra (DMSO-*d*_6_) in combination with ^1^H-^13^C HSQC experiments of **2** exhibited the presence of 31 carbons, including: one methyl CH_3_-31[(*δ_H_* 2.14 (3H s), (*δ_C_* 21.9 q)], nineteen methylenes grouped by interpretation of ^1^H-^1^H COSY and 1,1-ADEQUATE experiments into five spin systems, including: H_2_-2 to H_2_-6; H_2_-9 and H_2_-10; H_2_-13 to H_2_-17; H_2_-20 and H_2_-21; H_2_-24 to H_2_-28 and five quaternary carbonyl carbons C-8 (*δ_C_* 173.9 s), C-11 (*δ_C_* 174.7 s), C-19 (*δ_C_* 173.9 s), C-22 (*δ_C_* 174.9 s) and C-30 (*δ_C_* 173.8 s).

Seven hydrogen resonances lacked correlations in the ^1^H-^13^C HSQC spectrum of **2** and were therefore recognized as being located on either oxygen or nitrogen. From the results of a ^1^H-^15^N HSQC measurement made with **2** it was evident that four protons were bonded to nitrogen: comprising one NH_2_ group; NH_2_-1 and two NH groups; NH-12 [*δ_H_* 7.79 (1H, brs)] and NH-23 [*δ_H_* 7.79 (1H, brs)]. Also, from the ^1^H-^15^N HSQC and ^1^H-^15^N HMBC 2D-NMR spectra of **2** (Figure 3) it was possible to assign the resonance of each nitrogen: NH_2_-1 (*δ_N_* 31.4 t), NH-12 (*δ_N_* 116.2 d), NH-23 (*δ_N_* 116.2 d), N-7 (*δ_N_* 174.4 s), N-18 (*δ_N_* 174.4 s) and N-29 (*δ_N_* 175.6 s).

With all protons assigned to their directly bonded carbon and nitrogen atoms it was possible to deduce substructures. The connectivities between these substructures were established from key ^1^H-^13^C HMBC and ^1^H-^15^N HMBC correlations (Figure 3 and Appendix A). The positions of nitrogen in amide formation were confirmed by ^15^N-HMBC that showed correlation from H-3 to NH_2_-1; H-5 to N-7; H-10, H-13 and H-14 to NH-12; H-16 to N-18; H-21, H-24 and H-25 to NH-23 and H-27 and H-31 to N-29 (Appendix A). The 1,1-ADEQUATE experiment confirmed the correlations of ^1^H-^1^H COSY and partial substructures through its two bond correlations (Figure 3 and Appendix A). The 1,1-ADEQUATE is a technique used to obtain heteronuclear correlations similarly to ^1^H-^13^C HMBC. While correlation signals from HMBC do not separate ^2^*J*_CH_ from ^3^*J*_CH_, 1,1-adequate, which exclusively observes ^1^*J*_CH_ and ^2^*J*_CH_, and can be combined with ^1^H-^13^C HSQC to identify ^2^*J*_CH_. Interpretation of HSQC-TOCSY confirmed the spin systems via correlations from H-7, H-6, H-5, H-4, H-3 to C-2; H-9 to C-10; NH-12, H-14, H-15, H-16, H-17 to C-13; H-20 to C-21 and NH-23, H-25, H-26, H-27, H-28 to C-24 and confirmed the full structure of desferrioxamine B (**2**).

### 2.3. Genome Sequencing and Annotation


The whole genome sequencing reads of strain MT25^T^, generated using an Ion Torrent PGM instrument, 316v2 chips and Ian on PGM Hi-QTM View Sequencing Kit, were assembled using the Ion Torrent SPAdes plugin (v. 5.0.0.0) program (Life Technologies Limited, Paisley, UK). The size of whole genome sequence of the strain represented by 1170 contigs is 6,053,796 bp with a G + C content of 71.6%. Additional genomic features of the strain are shown in Table 2 according to GenBank NCBI prokaryotic genome annotation pipeline [24,25,26].

### 2.4. Phylogeny


The phylogenetic tree (Figure 4) based on almost complete 16S rRNA gene sequences shows that *Micromonospora* strain MT25^T^ belongs to a well-supported lineage together with the type strains of nine *Micromonospora* species. It is most closely related to *M. chalcea* DSM 43026^T^. With only 4 nucleotides difference within a 1437 sequence, the 16S rRNA sequences of these two strains are 99.7% identical. The 16S rRNA of strain MT25^T^ also shares a relatively high sequence identify with the *Micromonospora aurantiaca* [27], *Micromonospora marina* [28], *Micromonospora maritima* [29], *Micromonospora sediminicola* [30] and *Micromonospora tulbaghiae* [31,32] strains. The close relationship between these species is in a good agreement with the results from previous 16S rRNA gene sequence analyses [8,33]. The sequence similarities between the 16S rRNA sequences of strain MT25^T^ and the other *Micromonospora* strains range from 88.6 to 99.1%, which is equivalent to 13 to 20 nucleotide differences.

Greater confidence can be placed in the topology of phylogenetic trees based on whole genome sequences than on corresponding 16S rRNA gene trees, as the former are generated from millions, as opposed to hundreds, of unit characters [5]. The phylogenomic tree (Figure 5) shows that the strain MT25^T^ is most closely related to *M. tulbaghiae* DSM 45124^T^. In turn, these strains belong to a well-supported lineage which includes the *M. aurantiaca*, *M. chalcea*, *M. marina*, *M. maritima* and *M. sediminicola* strains together with the type strain of *M. humi* [34], all of these species belong to a distinct taxon, group 1a, highlighted in the genome-based classification of the genus *Micromonospora* generated by Carro et al. [8].

The recommended thresholds used to distinguish between closely related prokaryotic species based on average nucleotide identity (ANI) and digital DNA-DNA hybridization (dDDH) values are 95 to 96% [35,36] and 70% [36,37], respectively. Table 3 shows that the ANI and dDDH similarities between strain MT25^T^ and *M. aurantiaca* ATCC 27029^T^, *M. chalcea* DSM 43026^T^ and *M. marina* DSM 45555^T^, its three closest phylogenomic neighbors, are below the cut-off points used to assign closely related strains to the same species. The ANI and dDDH values also provide further evidence that strain MT25^T^ is most closely related to *M. tulbaghiae* DSM 45142^T^. However, the relationship between these strains is not clear-cut as they share a dDDH value below the 70% threshold and an ANI value at the borderline used to assign closely related strains to the same species. Conflicting results such as these are not unusual, as exemplified by studies on closely related *Micromonospora* and *Rhodococcus* species [33,38]. In such instances, ANI and dDDH similarities need to be interpreted with a level of flexibility and should also be seen within the context of other biological features, such ecological, genomic and phenotypic criteria [33,38,39]. Again, the use of a universal ANI threshold for the delineation of prokaryotic species has been questioned [40].

### 2.5. Species Assignment


It can be seen from Table 4 that strain MT25^T^ and *M. tulbaghiae* DSM 45142^T^, its closest phylogenomic neighbor, have phenotypic features in common though a range of other properties can be weighted to distinguish between them. Strain MT25^T^, unlike the *M. tulbaghiae* strain, grows at pH 6 and 10, reduces nitrate and shows much greater activity in the AP1-ZYM tests. In contrast, the *M. tulbaghiae* strain, unlike strain MT25^T^, grows at 4 °C, in the presence of 5% *w*/*v* sodium chloride, produces hydrogen sulfide and shows greater activity in the degradation tests. In addition, strain MT25^T^ produces sessile, rugose ornamental single spores on the substrate mycelium (Figure 1) whereas the *M. tulbaghiae* strain bears smooth, single spores borne on sporophores [31]. Further, strain MT25^T^ produces an orange as opposed to a brown substrate mycelium on yeast-malt extract agar though the colonies of both strains become dark brown/black on sporulation. The two strains also have different cellular sugar profiles as only strain MT25^T^ produces mannose. They can also be distinguished using a range of genomic features, notably genome size and G + C content. The genome size of strain MT25^T^ is 6.05 Mbp and its G + C content is 71.6%, whilst the corresponding figures for the *M. tulbaghiae* strain are 6.5 Mbp and 73.0%. Genome size and G + C content are considered to be conserved within species and can therefore represent useful taxonomic markers [5]. Inter-species variation in genomic G + C content does not usually exceed 1% [5,41].

In light of all of these data, it can be concluded that although strains MT25^T^ and *M. tulbaghiae* DSM 45142^T^ are close phylogenomic neighbors which can be distinguished using a combination of genomic and phenotypic properties, notably their genome sizes and G+C contents. It is, therefore, proposed that isolate MT25^T^ be considered as the type strain of a novel *Micromonospora* species that belongs to the phylogenomic group 1a, as designed by Carro et al. [8]. The name proposed for this species is *Micromonospora provocatoris* sp. nov.

### 2.6. Description of Micromonospora provocatoris sp. nov.

*Micromonospora provocatoris* (pro.vo.ca.to`ris. L. gen. n. *provocatoris*, of a challenger, referring to the Challenger Deep of the Mariana Trench, the source of the isolate), Aerobic, Gram-positive strain, non-acid-fast actinobacterium which forms nonmotile, single, sessile spores (0.8–0.9 µm) with rugose ornamentation on extensively branched substrate hyphae, but does not produce aerial hyphae. Colonies are orange on oatmeal agar eventually turning black on sporulation (Appendix A). Growth Occurs between pH 6.0 and 8.0, optimally at pH 7.0, from 10 °C to 37 °C, optimally at 28 °C and in the presence of 1% *w*/*v* sodium chloride. Aesculin is hydrolyzed and catalase produced. Degrades arbutin and L-tyrosine, but not starch or xylan. Furthermore, acid and alkaline phosphatases, α-chymotrypsin, cystine, leucine and valine arylamidases, esterase (C4), lipase esterase (C8), lipase (C14), *β*-galactosidase, *β*-glucosidase, naphthol-AS-BI-phosphohydrolase and trypsin are produced, but not α-fucosidase, *α*-galacturonidase, *β*-glucuronidase or *α*-mannosidase. The cell wall contains *meso*-A_2_pm, and the whole cell sugars are glucose, mannose, ribose and xylose. The predominant fatty acid is iso-C16:0 and the polar lipid profile contains diphosphatidylglycerol, phosphatidylethanolamine and phosphatidylinositol, a glycolipid and two unidentified phospholipids. The dDNA G + C content of the type and only strain is 71.6% and it is genome size 6.05 Mbp.

The type strain MT25^T^ (= NCIMB 15245^T^ = TISTR 2834^T^) was isolated from surface sediment from the Challenger Deep in the Mariana Trench of the Pacific Ocean. The accession numbers of the 16S rRNA gene sequence and that of the whole genome of the strain are AY894337 and QNTW00000000, respectively.

### 2.7. Specialised Metabolite-Biosynthetic Gene Clusters


Antibiotic and Secondary Metabolites Analysis Shell “AntiSMASH 6.0.0 0 alpha 1” [42] predicts natural products-biosynthetic gene clusters (NP-BGCs) that are based on the percentage of genes from the closest known bioclusters which share BLAST hits to the genome of the strains under consideration. Mining the draft genome of *M. provocatoris* MT25^T^ revealed the presence of ten known BGCs (Table 5). Two gene clusters were predicted to be responsible for the biosynthesis of siderophore desferrioxamine B, which was initially isolated from *Streptomyces* strain 1D38640 [43], and rhizomide A, which has antitumor and antimicrobial properties [44]. The other gene clusters found are likely to be involved with the biosynthesis of such products as phosphonoglycans, alkyl-*O*-dihydrogeranyl-methoxyhydroquinones [45], and the antibiotics kanamycin [46], brasilicardin A and frankiamicin [47,48]. Interestingly, two bioclusters belonging to two classes I lanthipeptides and a class III lanthipeptide lacked any homology thereby providing further evidence that NP-BGCs are discontinuously distributed in the genomes of *Micromonospora* taxa [8,9].

### 2.8. Genes Potentially Associated with Enviromental Stress


Stress-related genes detected in the genome of *Dermacoccus abyssi* strain MT1.1^T^, an isolate from the same sediment sample as the *M. provocatoris* strain, gave clues to how this piezotolerant strain became adapted to environmental conditions which prevail in sea-floor sediment of the Challenger Deep of the Mariana Trench [49]. In the present study, the genome of *M. provocatoris* strain MT25^T^ annotated using NCBI Genbank [24,25,26] pipeline was seen to harbor genes associated with a range of stress responses, notably ones linked with carbon starvation, cold shock response, high pressure, osmoregulation and oxidative stress (Appendix A), as was the case with the *D. abyssi* strain.

Deep-sea psychrophilic bacteria synthesize cold shock proteins essential for adaptation to low temperatures [50,51,52]. The genome of strain MT25^T^ contained genes predicted to encode cold shock proteins, as exemplified by genes *clpB* and *hscB* which are associated with the synthesis of ATP-dependent and Fe-S chaperones, respectively [52,53,54]. The genome also contains gene *deaD* encoding an RNA helicase involved in cold shock response and adaptation [55]. The strain has genes associated with the synthesis of branch-chain and long chain polysaturated fatty acids that are linked to membrane fluidity and functionality at low temperatures [49,52], including *fabF*, *fabG*, *fabH* and *fabI* genes which are responsible for the biosynthesis of *β*-ketoacyl-ACP synthase II, 3-oxoacyl-ACP reductase, ketoacyl-ACP synthase III, enoyl-ACP reductase and enoyl-ACP reductase, respectively (Appendix A). The synthesis of low-melting point branched-chain and/or polyunsaturated fatty acids (PUFAs) is crucial as it allows organisms in cold environments to maintain membrane fluidity in a liquid crystalline state thereby allowing organisms to resist freeze-thaw cycles at low temperatures [56,57]. Low temperatures reduce enzymatic activity leading to the generation of reactive oxygen species (ROS). The genome of strain MT25^T^ contains genes *sodN*, *trxA* and *trxB* predicted to encode products that offset the harmful effects of superoxide dismutase, thioredoxin and thioredoxin-disulfide reductase respectively.

Bacteria living in deep-sea habitats have developed ways of dealing with osmotic stress, notably by synthesizing osmoregulators, these are small organic molecules (compatible solutes) induced under hyperosmotic stress [58,59,60]. In this context, strain MT25^T^ contains genes predicted to be involved in the biosynthesis of compatible solutes, such as *opuA* gene, which regulates the uptake of glycine/betaine thereby contributing to osmotic stress responses [61,62]. Similarly, genes *asnO* and *ngg* are predicted to be involved in the production of osmoprotectant NAGGN (*n*-acetylglutaminylglutamine amide) that has an important role in counteracting osmotic stress in deep-sea environments. It is produced by many bacteria grown at high osmolarity bacteria, such as *Sinorhizobium meliloti* [63].

Another consequence of high pressure on bacteria is that the transport of compounds, such as amino acids, is reduced leading to upregulation of transported molecules [64]. Genes associated with the production of different types of ABC transporter permeases were detected in strain MT25^T^ including branched-chain amino acid permeases that are upregulated at high pressure [65]. In addition, the genome of strain MT25^T^ contains pressure sensing and pressure adaptation genes, as illustrated by *cycD*, *mdh* and *asd* genes, which are linked to the production of a thiol reductant ABC exporter subunit, malate dehydrogenase and aspartate semialdehyde dehydrogenase, respectively. Similarly, *secD* and *secF* are predicted to encode protein translocase subunits and *secG* preprotein translocase unit [65,66], as shown in Appendix A.

Bacteria able to grow in nutrient-limiting conditions need to store carbon compounds like glycogen [67]. In this respect, it is interesting that strain MT25^T^ contains a gene, *gigA*, which is predicted to encode glycogen synthase and another gene, *gigx*, which is linked with the production of a glycogen debranching enzyme responsible for the breakdown of this storage molecule. Furthermore, the strain has the capacity to produce carbonic anhydrase proteins which are required for fixation of carbon dioxide [65,68] thereby suggesting that its potential to grow as a lithoautotroph. This discovery provides further evidence that filamentous actinobacteria in carbon-limiting, extreme biomes are capable of adopting a lithoautotrophic lifestyle, as shown by the type strains of novel *Blastococcus*, *Geodermatophilus* and *Modestobacter* species [69,70,71,72,73].

Micromonosporae can grow under aerobic and microaerophilic conditions. Their ability to tolerate low oxygen tensions indicates an ability to grow in oxygen depleted biomes, such as lake and river sediments and soil prone to flooding [74,75]. Genome mining of strain MT25^T^ revealed many putative genes predicted to encode terminal oxidases involved in aerobic respiration, as witnessed by the *cydB* gene encoding cytochrome *d* ubiquinol oxidase subunit II, and genes *ctad* and *coxb* expressing cytochrome *c* oxidase subunits I and II, respectively. Several terminal dehydrogenase and reductase encoding genes involved in respiratory chains were detected, including ones predicted to express arsenate reductase *arsc* and ferredoxin reductase. Multiple genes predicted to encode succinate dehydrogenase used as electron donors under low oxygen conditions were also detected in the genome of strain MT25^T^. Further support for the ability of the strain to adapt to different oxygen levels reflects its capacity to form cytochrome oxidase complexes that have different affinities for oxygen. Biological adaptations such as these may account for the presence of micromonosporae (including verrucosisporae) in marine habitats, including deep-sea sediments [2,3,76].

## 3. Materials and Methods

### 3.1. Microorganism


*Micromonospora* strain MT25^T^ was isolated from Mariana Trench sediment, sample no. 281, collected at a depth of 10,898 m (Challenger Deep; 11°19′911′′ N; 142°12′372′′ E) by the remotely operated submersible Kaiko, using a sterilized mud sampler, on 21 May 1998, during dive number 74. The sample was transported to the UK in an insulated container at 4 °C and stored at −20 °C until examined for actinobacteria. The test strain was isolated, purified and maintained using procedures described by Pathom-aree et al. [19]. *M. tulbaghiae* DSM 45142^T^ was maintained under the same conditions.

### 3.2. General Experimental Procedures


General Experimental Procedures. ^1^H, ^13^C, ^15^N NMR experiments were recorded on a Bruker Avance 600 MHz NMR spectrometer AVANCE III HD (Billerica, MA, USA) equipped with a cryoprobe, in DMSO-*d*_6_. Low resolution electrospray mass spectra were obtained using a Perseptive Biosystems Mariner LC-MS (PerSeptive Biosystems, Framingham, MA, USA), and high-resolution mass data were generated on Finnigan MAT 900 XLT (Thermo-Finnigan, San Jose, CA, USA). HPLC separations were carried out using a Phenomenex reversed-phase (C_18_, 10 Å × 10 mm × 250 mm) column and an Agilent 1100 series gradient pump and monitored using an Agilent DAD G1315B variable-wavelength UV detector (Agilent Technologies, Waldbronn, Germany).

### 3.3. Fermentation Conditions


For the first-stage seed preparation, an agar grown culture of strain MT25^T^, was inoculated into 10 mL of GYE medium (4.0 g glucose, 4.0 g yeast extract, agar 15 g, distilled H_2_O 1 L, pH 7.0). After 5 days incubation at 28 °C, with agitation, the first stage culture was used to inoculate the production fermentation, using ISP2 broth (yeast extract 4 g, malt extract 10 g, glucose 4 g, CaCO_3_ 2 g, distilled H_2_O 1 L, pH 7.3). The fermentation was incubated at 28 °C, with agitation, and the biomass was harvested on the seventh day. All media components were purchased from Sigma-Aldrich (St. Louis, MO, USA).

### 3.4. Isolation and Purification of Secondary Metabolites


Harvested fermentation broth (6 L) was centrifuged at 3000 rpm for 20 min, and the HP20 resin together with the cell mass was washed with distilled water then extracted with MeOH (3 × 500 mL). The MeOH extracts were combined and concentrated under reduced pressure to yield 6.39 g solid extract. The extract was suspended in 250 mL of MeOH and then partitioned with *n*-hexane (3 × 250 mL). The remaining MeOH solubles were the subject of further purification by Sephadex LH-20 column chromatography (CH_2_Cl_2_/MeOH 1:1) to yield 3 fractions. Final purification was achieved using reversed–phase HPLC (C_18_, 10 µm, 10 mm × 250 mm), employing gradient elution from 0–90% CH_3_CN/H_2_O containing 0.01% TFA over 40 min for fraction A (23 mg) to give compound **1** (16.2 mg) and fraction B (27 mg) and to give compound **2** (9.4 mg).

Compound (**1**): white amorphous powder, 16.2 mg; ${\left[\alpha \right]}_{D}^{25}$ = −47 (*c* 0.1, MeOH); IR ν_max_: 3408, 3326, 3316, 3274, 3230, 3202, 1670, 1660, 1655, 1647, 1445, 1237 cm^−1^; LRESIMS *m*/*z* 338.10 [M + Na]^+^; HRESIMS *m*/*z* 338.143098 [M + Na]^+^ (calcd for C_12_H_21_N_5_O_5_Na, 338.144046, Δ = −2.8 ppm).; ^1^H and ^13^C NMR data (DMSO-*d_6_*), see Table 1.

Compound (**2**): colorless amorphous substance, 9.4 mg; IR ν_max_: 3315, 3090, 2860, 1625, 1560, 1460, 1270, 1225, 1190 cm^−1^; HRESIMS *m*/*z* 561.3577 [M + H]^+^ (calcd for C_25_H_49_N_6_O_8_, 561.3592, Δ = −2.6 ppm).; ^1^H and ^13^C NMR data (DMSO-*d_6_*), see Table 1.

### 3.5. Phylogeny


An almost complete 16S rRNA gene sequence (1437 nucleotides) (Genbank accession number AY894337) was taken directly from the draft genome of the isolate using the ContEst16S tool from the EzBioCloud webserver (https://www.ezbiocloud.net/tools/contest16s, accessed on 1 June 2018) [77]. The sequence was aligned with corresponding sequences of the most closely related type strains of *Micromonospora* species drawn from the EzBioCloud webserver [78] using MUSCLE software (Version No. 3.8.31, drive5, Berkeley, CA, USA) [79]. Pairwise sequence similarities were generated using the single gene tree option from the Genome-to-Genome Distance calculator (GGDC) webserver [37,80] and phylogenetic trees inferred using the maximum-likelihood [81], maximum-parsimony [82] and neighbor-joining [83] algorithms. A ML (maximum likelihood) tree was generated from alignments with RAxML (Randomized Axelerated Maximum Likelihood) [84] using rapid bootstrapping with the auto Maximum-Relative-Error (MRE) criterion [85] and a MP tree inferred from alignments with the tree analysis using the New Technology (TNT) program [86] with 1000 bootstraps together with tree-bisection-and-reconnection branch swapping and ten random sequence addition replicates. The sequences were checked for computational bias using the x_2_ test taken from PAUP * (Phylogenetic analysis using parsimony) [87]. The trees were evaluated using bootstrap analyses based on 1000 replicates [88] from the MEGA X software package (Version No. 10.0.5, MEGA development team, State College, PA, USA) [89] and the two-parameter model of Jukes and Cantor, 1969 [90]. The 16S rRNA gene sequence of *Catellatospora citrea* IFO 14495^T^ (D85477) was used to root the tree.

### 3.6. Phenotypic Characterisation


The isolate was examined for a broad range of phenotype properties known to be of value in *Micromonospora* systematics [10,16]. Standard chromatographic procedures were used to detect isomers of diaminopimelic acid [91], whole-organism sugars [92] and polar lipids [93,94], using freeze dried biomass harvested from yeast extract-malt extract broth cultures (International Streptomyces Project [ISP] medium 2) [95]. Similarly, cellular fatty acids extracted from the isolate were methylated and analyzed using the Sherlock Microbial Identification (MIDI) system and the resultant peaks identified using the ACTINO 6 database [96].

Cultural and morphological properties of the isolate were recorded following growth on oatmeal agar (ISP medium 3) [95]. Growth from the oatmeal agar plate was examined for micromorphological traits using a scanning electron microscope (Tescan Vega 3, LMU instrument, Fuveau, France) and the protocol described by O’Donnell et al. [97]. The enzymatic profiles of strain MT25^T^ and M. tulbaghiae DSM 45142^T^ were determined using AP1-ZYM strips (bioMérieux) by following the instructions of the manufacture. Similarly, biochemical, degradation, physiological and staining properties were acquired using media and methods described by Williams et al. [98]. The ability of strain MT25^T^ to grow under different temperature and pH regimes and in the presence of various concentrations of sodium chloride were recorded on ISP2 agar as the basal medium; the pH values were determined using phosphate buffers. All of these tests were carried out using a standard inoculum of spores and mycelial fragments equivalent to 5.0 on the McFarland scale [99].

### 3.7. Whole-Genome Sequencing

#### 3.7.1. DNA Extraction and Genome Sequencing

Genomic DNA was extracted from wet biomass of a single colony of strain MT25^T^ following growth on yeast extract-malt extract agar for 7 days at 28 °C [95], using the modified CTAB method [100]. The sequence library was prepared using a NEB Next Fast DNA Fragmentation and Library Preparation Kit for an Ion Torrent (New England Biolabs, Hitchin, UK).

Briefly, the DNA sample (0.5 μg) was subjected to enzymatic fragmentation, end repaired and ligated to A1 and P2 adapters, followed by extraction of 490–500 bp fragments and PCR amplification. The PCR products were analyzed using a High Sensitivity DNA kit and BioAnalyser 2100 (Agilent Technologies LDA).

(UK Limited, Cheshire, UK). AMPure XP beads (Beckman Coulter, Brea, CA, USA) were used for DNA purification according to the protocol. The library was diluted to give a final concentration of 25 pM, and a template was prepared using an Ion PGM Hi-Q™ (Life Technologies Limited, Paisley, UK) View OT2 Kit and IonTorrent One Touch system OT2. The recovery of positive Ion Sphere Particles was achieved using the One Touch ES enrichment system. The sequencing reaction was conducted using an Ion PGM Hi-Q^TM^ View Sequencing Kit, 316v2 chips and an IonTorrent PGM instrument with 850 sequencing flows, according to manufacturer’s instructions (Life Technologies Limited, Paisley, UK), required for 400 nt read lengths.

#### 3.7.2. Annotation of Genome and Bioinformatics

The sequencing reads were mapped onto reference genome sequences using CLC Genomics Workbench software (GWB, ver. 7.5, QIAGEN, LLC, Germantown, MD, USA). The reads were assembled using SPAdes v. 5.0.0.0 plugin (LifeTechnologies, Thermo Fisher Scientific, UK). The annotation of the genomic sequence was performed via NCBI GenBank annotation pipeline [24,101].

#### 3.7.3. Detection of the Gene Clusters


The whole genome sequence of strain MT25^T^ was mined using AntiSMASH 6.0.0 alpha 1 (“Antibiotic and Secondary Metabolites Analysis Shell”) [42] to detect biosynthetic gene clusters. The NCBI [24,25,26] GenBank annotation pipeline was used to detect the genes and proteins associated with bacterial adaptation.

#### 3.7.4. GenBank Accession Number


This Whole Genome Shotgun sequence has been deposited at DDBJ/ENA/GenBank under accession number NZ_QNTW00000000. The version described in this paper is NZ_QNTW00000000.1.

### 3.8. Comparison of Genomes


The draft genome sequence of strain MT25^T^ was compared with corresponding sequences of the type strains of closely related *Micromonospora* strains, as shown in the phylogenomic analyses. A ML phylogenomic tree inferred using the codon tree option in the PATRIC webserver [102], based on aligned amino acids and nucleotides derived from 704 single copy core genes in the genome dataset matched against the PATRIC PGFams database (http://www.patricbrc.org, accessed on 10 July 2018), was generated using the RAxML algorithm [84]. Average nucleotide identity (ortho ANI) [103] and digital DNA-DNA hybridization [38] values were determined between the isolate and the type strains of *M. aurantiaca*, *M. chalcea*, *M. marina* and *M. tulbaghiae*, its closest phylogenomic neighbors.

## 4. Conclusions

*Micromonospora* strain MT25^T^, an isolate recovered from sediment taken from the Mariana Trench in the Pacific Ocean, was shown to be most closely related to the type strain *M. tulbaghiae* following a genome-based classification. Characterization of strain MT25^T^ using a range of methods suggests that it belongs to a new *Micromonospora* species, which we name as *Micromonospora provocatoris* sp. nov. An associated bioassay-guided study together with structural analyses showed that the isolate has a potential to synthesize two major metabolites, *n*-acetylglutaminyl glutamine amide and desferrioxamine B. In line with previous studies on micromonosporae isolated from extreme habitats, strain MT25^T^ had a relatively large genome containing genes likely to be involved in the biosynthesis of novel natural products. Bioinformatic analyses of the genome of the *M. provoactoris* strain revealed a broad range of stress-related genes relevant to its survival in deep-sea sediments.

## Figures and Tables

**Figure 1 marinedrugs-19-00243-f001:**
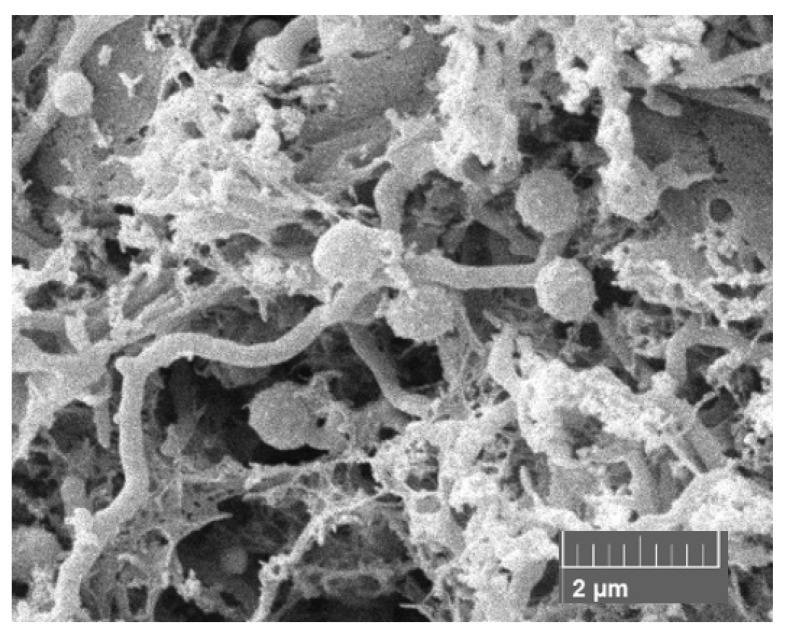
Scanning electron micrograph of *Micromonospora* strain MT25^T^ (× 2.4 k), showing the presence of single sessile spores with the rugose surfaces borne on the substrate mycelium following growth on oatmeal agar for 7 days at 28 °C.

**Figure 2 marinedrugs-19-00243-f002:**
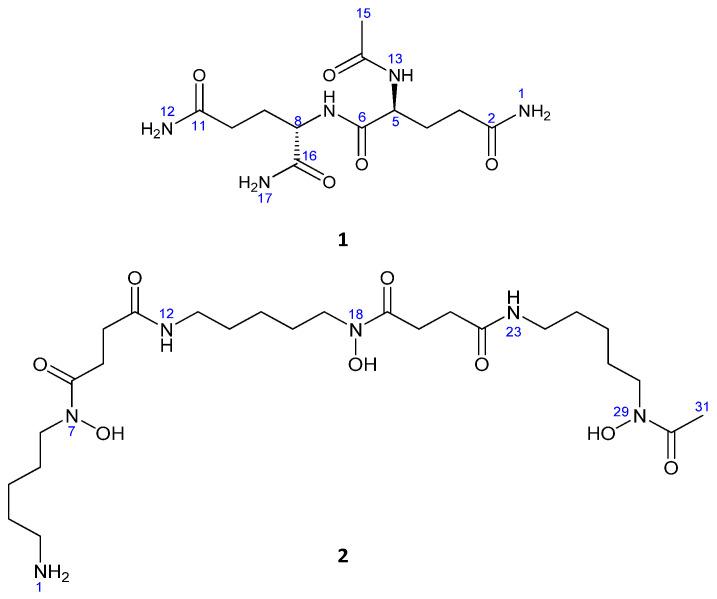
Chemical structures of the isolated compounds.

**Figure 3 marinedrugs-19-00243-f003:**
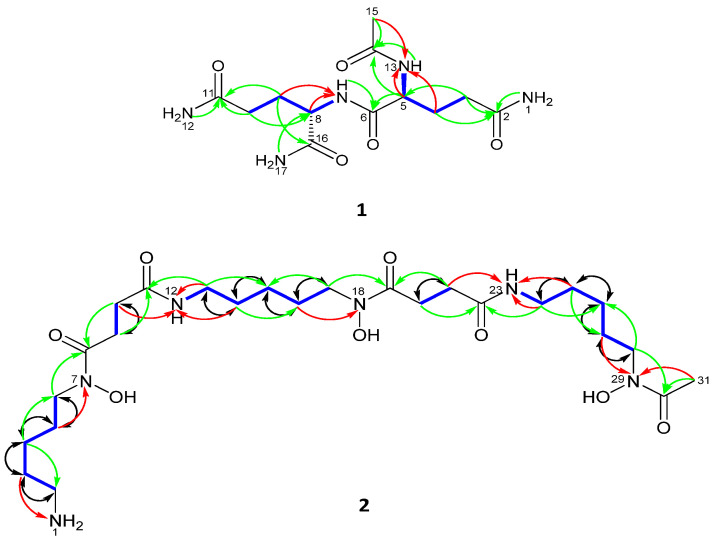
Selected COSY (blue), ^1^H-^13^C HMBC (H→C) (green), ^1^H-^15^N HMBC (H→N) (red) and 1,1-ADEQUATE (H→C) (black) correlations.

**Figure 4 marinedrugs-19-00243-f004:**
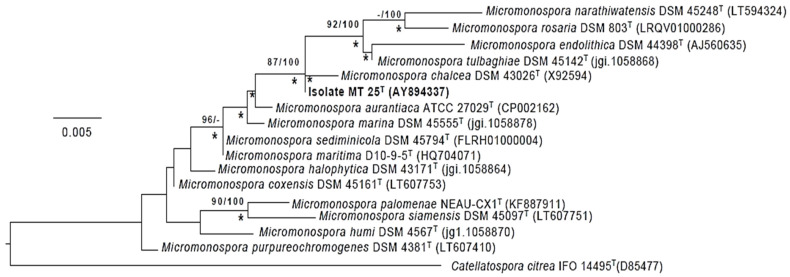
Maximum-likelihood tree based on almost complete 16S rRNA gene sequences generated using the GTR-GAMMA model showing relationships between isolate MT25^T^ and the type strains of the closest phylogenetic neighbors. Asterisks indicate branches of the tree that were also formed using the maximum-parsimony and neighbor-forming algorithms. The numbers at the nodes are bootstrap support values greater than 60% for ML (left) and MP (right). The root of the tree was established using *Catellatospora citrea* IFO 14495^T^. The scale bar indicates 0.005 substitutions per nucleotide position.

**Figure 5 marinedrugs-19-00243-f005:**
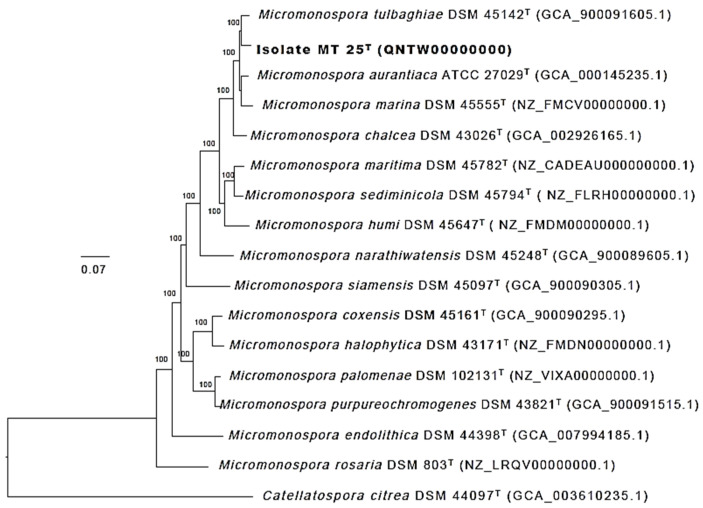
Maximum-likelihood phylogenomic tree based on 704 single copy core genes showing relationships between isolate MT25^T^ and closely related type strains of *Micromonospora* species. Numbers at the nodes are bootstrap support values based on 100 replicates. GenBank accession numbers are shown in parentheses. The scale bar indicates 0.07 substitutions per nucleotide position. The tree is rooted using the type strain of *Catellatospora citrea*.

**Table 1 marinedrugs-19-00243-t001:** ^1^H (600 MHz), ^13^C NMR (150 MHz) and ^15^N (600 MHz) data (in DMSO-*d*_6_) for compounds **1** and **2**.

No.	1			2		
	δ_C_, Mult.	δ_N_, Mult.	δ_H_, Mult (*J* in Hz)	δ_C_, Mult.	δ_N_, Mult.	δ_H_, Mult (*J* in Hz)
1	-	108.5, NH_2_	**a.** 7.28, brs **b.** 6.75, brs		31.4 t	
2	173.8, C	-	-	41.4 t		2.76 (t,8.0)
3	31.4, CH_2_	-	2.06, m	29.2 t		1.51 m
4	27.2, CH_2_	-	**a.** 1.89, m **b.** 1.74, m	25.4, t		1.38 m
5	52.7, CH	-	4.15, m	28.3 t		1.51 m
6	171.4, C	-	-	49.3 t		3.46 m
7	-	117.3, NH	7.97, d (7.8)	-	174.4, s	9.68 brs
8	52.1, CH	-	4.12, m	173.9 s		-
9	27.8, CH_2_	-	**a.** 1.87, m **b.** 1.67, m	30.0 t		2.58 m
10	31.6, CH_2_	-	2.08, m	31.3 t		2.27 m
11	173.8, C	-	-	173.9 s		-
12	-	108.6, NH_2_	**a.** 7.28, brs **b.** 6.75, brs	-	116.2 d	7.79 brs
13	-	123.2, NH	8.11, d (7.8)	41.0 t		3.00 (q, 8.0)
14	169.7, C	-	-	31.4 t		1.38 m
15	22.6, CH_3_	-	1.86, s	26.0 t		1.26 m
16	173.3, C	-	-	28.6 t		1.51 m
17	-	104.8, NH_2_	**a.** 7.27, brs **b.** 7.05, brs	49.4 t		3.46 m
18					174.4 s	9.67 brs
19				173.9 s		-
20				30.1 t		2.58 m
21				32.4 t		2.27 m
22				173.9 s		-
23					116.2 d	7.79 brs
24				41.0 t		3.00 (q, 8.0)
25				31.4 t		1.38 m
26				26.0 t		1.26 m
27				28.6 t		1.51 m
28				49.6 t		3.46 m
29				-	175.6 s	9.63 brs
30				173.5 s		-
31				22.9 q		1.96 s

**Table 2 marinedrugs-19-00243-t002:** Genomic features of *Micromonospora* strain MT25^T^.

Features	Strain MT25^T^
Assembly size, bp	6,053,796
No. of contigs	1170
G + C (%)	71.6
Fold coverage	39.94×
N50	8214
L50	203
Genes	6643
CDs	6573
Pseudo genes	2188
Protein encoding genes	4385
rRNA	8
tRNA	59
ncRNAs	3
Accession No.	NZ_QNTW00000000
Assembly method	SPAdes v. 5.0.0.0

**Table 3 marinedrugs-19-00243-t003:** Average nucleotide identity (ANI) and digital DNA-DNA hybridization (dDDH) similarities between *Micromonospora* strain MT25^T^ and its closest phylogenomic neighbors.

Phylogenomic Neighbors	Similarity ANI	Values (%) dDDH
*M. aurantiaca* ATCC 27029^T^	95.2	62.7
*M. chalcea* DSM 43026^T^	93.5	53.0
*M. marina* DSM 45555^T^	94.6	58.6
*M. tulbaghiae* DSM 45142^T^	96.0	68.1

**Table 4 marinedrugs-19-00243-t004:** Phenotypic properties that distinguish isolate MT25^T^ from *M. tulbaghiae* DSM 45142^T^.

Characteristics	Strain MT25^T^	*M. tulbaghiae* DSM 45142^T^
**Morphology:**		
Spores borne on sporophores	-	+
Spore ornamentation	Rugose	Smooth
Substrate mycelial color on yeast extract-malt extract agar	Orange	Brown
**AP1-ZYM tests:**		
Acid and alkaline phosphatases, β-glucosidase, lipase (C14), naphthol-AS-BI-phosphohydrolase	+	-
α-galacosidase, β-glucoronidase, N-acetyl-β-glucosaminidase	-	+
**Biochemical tests:**		
H_2_S production	-	+
Nitrate reduction	+	-
**Degradation tests:**		
L-tyrosine	+	-
Casein	-	+
Gelatin	-	+
Starch	-	+
Tween-80	-	+
**Tolerance tests:**		
Growth at 4 °C	-	+
Growth at pH 6.0 and pH 10	+	-
Growth in presence of 5% *w/v* NaCl	-	+
**Chemotaxonomy:**		
Major whole-organism sugars	Glucose, mannose, ribose and xylose	Glucose, ribose and xylose

+, positive results; -, negative results. * Data for the biochemical, chemotaxonomic, tolerance and morphological properties on the *M. tulbaghiae* DSM 45142^T^ were taken from Kirby and Meyer (2010) [31]. Each strain grew from 10 to 37 °C, from pH 7 to 9, and were positive for α-chymotrypsin, cystine and valine arylamidases, esterase (C4), esterase lipase (C8), β-galactosidase and trypsin, but negative for α-fucosidase, α-mannosidase and *β*-glucoronidase. Neither strain degraded xylan.

**Table 5 marinedrugs-19-00243-t005:** Identity of predicted natural product biosynthetic gene clusters using antiSMASH 6.0.0 alpha 1.

BGCs	No.	Nucleotide (nt) bp	Type	Accession Number	Homologue	Accession Number	Identity
Siderophore	1	6963	Desferrioxamine E	QNTW01000257	Desferrioxamine EBGC from *Streptomyces* sp. ID38640	MG459167.1	100%
T2PKS *	1	3695	Frankiamicin	QNTW01000523	Frankiamicin BGC from *Frankia* sp. EAN1pec	CP000820.1	28%
Terpene	1	20066	Isorenieratene	QNTW01000028	Isorenieratene BGC from *Streptomyces griseus subsp. griseus NBRC 13350*	AP009493.1	28%
Terpene	1	11057	Phosphonoglycans	QNTW01000118	Phosphonoglycans BGC from *Glycomyces* sp. NRRL B-16210	KJ125437.1	3%
Oligosaccharides	1		Brasilicardin A		Brasilicardin A BGC from *Nocardia terpenica* IFM 0406	KV411304.1	23%
NRPS ***	1	10526	Rhizomide (A-C)	QNTW01000131	Rhizomide A BGC from *Paraburkholderia rhizoxinica* HKI 454	NC_014718.1	100%
T3PKS **	1	12,601	Alkyl-*O*-dihydrogeranyl-methoxyhydroquinones	QNTW01000093	alkyl-O-dihydrogeranyl-methoxyhydroquinones biosynthetic gene cluster from *Actinoplanes missouriensis* 431	AP012319.1	28%
Lanthipeptide-class-i	1	26,371	Kanamycin	QNTW01000003	kanamycin biosynthetic gene cluster from *Streptomyces kanamyceticus*	AB254080.1	1%
Lanthipeptide-class-i	1	18,770	No match found	QNTW01000004	-	-	-
Lanthipeptide-class-iii	1	7750	No match found	QNTW01000229	-	-	-

* Type II and III PKS cluster, ** Type III PKS cluster and *** Non-ribosomal peptide synthetase cluster.

## Data Availability

The article contains all the data produced in this study.

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
