# Peer review of "Biotechnological and Ecological Potential of Micromonospora provocatoris sp. nov., a Gifted Strain Isolated from the Challenger Deep of the Mariana Trench"

_marinedrugs, 2021, doi:10.3390/md19050243_

Round 1

Reviewer 1 Report

Abdel-Mageed et al. discuss the isolation, genomic as well as chemical characterization of two metabolites from a Micromonospora strain, isolate MT25T,  recovered from a sediment from Mariana Trench. In the study, they undertook several approaches to phylogenetically characterised this strain by sequencing its genome. Additionally, they evaluated some phenotypic features and proposed this strain to be classified as Micromonospora provocatoris sp. nov. By probing the genome, they identify biosynthetic  clusters as well as several genes needed for adaptation to deep sea environment.

References needed:

Contreras-Castro L, Maldonado LA, Quintana ET, Carro L, Klenk H-P. 2019. Genomic insight into three marine Micromonospora sp. strains from the Gulf of California. Microbiol Resour Announc 8:e01673-18.

Introduction:

Line 56-62: Please rephrase this sentence into two separate sentences to allow clarity.

Results and Discussion:

Line 89: This is unclear..."Like the other micromonosporae, the strain contains complex mixtures". Is there a figure which specifically shows this results? If yes, please specify.

Line 183: The genome was assembled into 1,170 contigs with  N50 of 8,214. This indicates to some degree of fragmentation that the authors need to mention.

Line 183: Table 2 provides the genomic features of Micromonospora strain MT25. It will be helpful to have a comparison of other closely related strain as a way of knowing different and similarities between genomic content.

Line 190: I am a bit concern about how only 4 nucleotides with the 1437 nucleotides 16s rRNA  of strain MT25T differs from the  closely related to M. chalcea 190 DSM 43026T. What was the sequencing accuracy of Ion Torrent PGM?

Line 190-200:  To ascertain the 16s rRNA and whole genome phylogenetics, the authors perform such analysis with limited Micromonospora strains. Although it is helpful to find the identify of strain MT25T, adding recently available uncharacterised Micromonospora  strains will also confirm whether strain MT25T is indeed novel or not.

Line 272: The strain is described as Micromonospora provocatoris; however the submitted assembly QNTW00000000, shows Micromonospora chalcea . Please confirm this.

Line 294: Using Anti-SMASh several BGCs were identified. It will be helpful if the authors could discuss further if any of the isolated and characterised metabolites can be linked to such BGCs. T

Materials and methods:

Line 426:  It will be helpful if the authors can provide their 16S rRNA gene sequence alignment as a supplementary file.

Figures:

Figure 1: Please correct the typo "sessiile"

Fig. S1: Please correct the typo "Twoo"

Fig. S2: This figure shows the lipid patterns of Micromonospora provoctaris MT25T. It is unclear whether some standards were used here to confirm the identity of these polar lipids. Please clarify.

Reviewer 2 Report

I read with interest the manuscript. 
This manuscript will be suitable for publishing once the minor changes indicated below are made.

Minor comment

- In Lane 3, 'sp. nov.' is in italic in the Title. Check it, please.

- In Fig. 2, there is an error in the numbering of compound 1. No. 8 should be modified to no. 7. 

- 2D NMR data such as COSY, HSQC, and HMBC use 1H-NMR to determine whether carbon or nitrogen is connected or bonded. In Fig. 3, the arrows indicating H-C or H-N correlation are reversed. This part needs to be corrected. And supplementary table S1 and S2 also should be revised. Don't use(paste) a figure of table.

- In the text, all NMR data are described for 600 MHz NMR. Therefore, it appears that the Varian Unity INOVA 400 MHz spectrometer described in Lane 391-392 would not be necessary. 

- The resolution of Figs 4 and 5 is poor. Replace it with a high-resolution picture. 
